# Global Maritime Container Shipping Networks 1969–1981: Emergence of Container Shipping and Reopening of the Suez Canal

Tomohiro Saito [1], Ryuichi Shibasaki [2,*], Shinsuke Murakami [3], Kenmei Tsubota [4] and Takuma Matsuda [5]

1 Department of Systems Innovation, School of Engineering, The University of Tokyo, Tokyo 113-8656, Japan; tomopiko0125@gmail.com
2 Resilience Engineering Research Center, School of Engineering, The University of Tokyo, Tokyo 113-8656, Japan
3 Department of Technology Management for Innovation, School of Engineering, The University of Tokyo, Tokyo 113-8656, Japan; smurakam@tmi.t.u-tokyo.ac.jp
4 Department of Regional Development Studies, Faculty of Global and Regional Studies, Toyo University, Tokyo 112-8606, Japan; kenmei.tsubota@gmail.com
5 Faculty of Commerce, Takushoku University, Tokyo 112-8585, Japan; tmatsuda@ner.takushoku-u.ac.jp
* Correspondence: shibasaki@tmi.t.u-tokyo.ac.jp; Tel.: +81-3-5841-6546

**Abstract:** This study applied graph theory to conduct an empirical analysis of the evolution of global maritime container shipping networks, mainly focusing on the 1970s. In addition to analyzing the change in overall structures of the networks over the long term (from the 1970s to the present) and midterm (in the 1970s), the authors examined the changes in the container shipping networks before and after the reopening of the Suez Canal in 1975. As a result, it was confirmed that the initial single polar network structure, in which New York and other North American ports were placed at the center, changed to a multipolar structure, finally forming a hub-and-spoke structure. Subsequently, the authors confirmed discontinuous changes in inter-regional density from 1975 to 1976 caused by an increase in the average number of ports of call in 1976, because the recession caused by the first oil crisis in 1973 decreased the maritime container shipping demand, and the reopening of the Suez Canal caused a surplus of containerships. This study would contribute to accumulating empirical knowledge on the vulnerability analysis of the present and future maritime container shipping networks.

**Keywords:** maritime container shipping (MCS); network analysis; Suez Canal (SC); weighted network; graph theory; 1970s; liner service (LS)

## 1. Introduction

Maritime shipping is one of the oldest means of transport. It has changed form, such as vessel type and cargo handling, but is still an important means of cargo transport. Among them, maritime container shipping (MCS), which was introduced in the middle of the 20th century, rapidly gained an important position. Since its emergence, MCS has continued to expand, except during periods of recession, and constitutes an integral part of the current globalized world economy. However, during its development, MCS has experienced many obstacles, including the closure of the Suez Canal (SC), two oil shocks, pirate attacks, economic downturns, and damage or shutdown of individual ports because of accidents, natural disasters, and strikes, but has overcome them by changing the network structure flexibly in some instances. Among them, the closure of the SC from 1967 to 1975 was considered to have a significant impact on maritime shipping, including MCS, because the SC is a key infrastructure located along the trunk routes connecting Europe and East Asia (EA). The year 2021 will record that the SC was closed in March owing to the grounding of a containership (*Ever Given*). Although the closure of the SC in 2021 lasted only around a

week, many impacts on global logistics were argued in the world's mass media. Therefore, discussing the impact of the closure of the SC until 1975 would provide useful lessons to the present global economies. In recent studies on MCS, network science techniques have been applied to assess its vulnerability and to understand its structural features. However, owing to a lack of available data, these studies mainly focused on the post-1970s period. In other words, changes in the network structure in the era of the emergence of MCS, and the impact of the reopening of the SC in 1975, have not been sufficiently verified. Therefore, there is a lack of sufficient empirical knowledge on these events.

This study analyzes global MCS networks in the 1970s, focusing mainly on the emergence of MCS and the reopening of the SC. We use a data source (Recent World Container Services (最近の世界コンテナ船就航状況) before 1977; World Container Fleets and Their Services (世界のコンテナ船隊および就航状況) after 1978) compiled by Nippon Yusen Kaisha (hereafter, the NYK report) for developing liner service (LS) network data of global MCS, mainly in the 1970s, and apply graph theory. Through the analyses, we reveal the developmental process of MCS in the 1970s and the changes of the network structures after the reopening of the SC.

The remainder of this paper is organized as follows: Section 2 reviews the relevant literature. Section 3 describes the data, models, and methods used in this study. Section 4 confirms the entire history of the development of the global MCS network since its emergence of MCS in the 1970s to 2016 to give a broader perspective to place our analysis. Section 5 focuses on the detailed changes in the era of the emergence of MCS, mainly in the 1970s, and Section 6 assesses the impact of the reopening of the SC in 1975 on the entire MCS. Finally, Section 7 summarizes the conclusions and future perspectives.

## 2. Literature

Transport network analysis based on graph theory is particularly common in the field of air and land transport. Research on the comprehensive network analysis of maritime shipping has also accumulated in recent years. Liu et al. (2018) [1] applied a weighted ego network analysis to visualize the spatial heterogeneity of maritime networks at the global and local levels. Fang et al. (2018) [2] proposed an automatic identification system (AIS)-based approach to understanding maritime network dynamics before and after international events, namely, military conflict between India and Pakistan, economic sanctions on Iran, and government elections in Sri Lanka. Toriumi and Watanabe (2012) [3] analyzed vessels sailing in the region where piracy and armed robbery incidents occurred using piracy data provided by the International Maritime Bureau and Lloyd's data on vessel movement. Wang et al. (2017) [4] analyzed the container service network of liner shipping companies in 2004, 2009, and 2013 between Taiwan and mainland China using a weighted network. Yu et al. (2019) [5] analyzed the tanker network structure and predicted flow changes by oil price variations using a systems-based approach to construct a maritime transportation network based on trajectory data.

Moreover, regarding the network analysis focusing on the characteristics of the present global MCS network, Hu and Zhu (2009) [6] analyzed the LS network in 2006 and calculated ports with good connectivity by setting edges between all ports in the same LS. Ducruet and Notteboom (2012) [7] further extended Hu and Zhu (2009) [6] using Lloyd's data on the movement of containerships between ports in 1996 and 2006 and compared the network structures at each time point. Ducruet et al. (2010) [8] analyzed Northeast Asian liner networks in 1996 and 2006 by hierarchical clustering with indicators such as hub dependence, degree distribution, and foreland diversity index. Pan et al. (2019) [9] applied the eigenvalue decomposition method to the LS network of the seven largest MCS companies in 2005. Cheung et al. (2020) [10] also used eigenvector centrality and frequency weights for the analysis. Kawasaki et al. (2019) [11] applied the proximity centrality method to intra-Asian LS data in 2016. In most of these analyses, the topological aspects of the MCS network were identified by connecting the ports in a sequential order that vessels called at in each LS. However, because the MCS network is composed of multiple LSs with

multiple port callings, in-service container movements and movements via transshipment between LSs should be distinguished. Therefore, this study considers not only a graph of direct linkages (GDL) by setting edges along the shipping movements but also a graph of all linkages (GAL) in the LS, similar to Hu and Zhu (2009) [6], Ducruet et al. (2010) [12], and Ducruet and Notteboom (2012) [7].

Some studies have focused on network vulnerabilities. For example, Stergiopoulos et al. (2018) [13] applied critical infrastructure dependency modeling to the positioning data of containerships from 2015 to 2017 obtained from the AIS. They evaluated the dependence between ports on vessel movement routes and identified routes and ports with high risk. Toriumi and Takashima (2012) [14] estimated the importance of global chokepoints, including the Bosporus, the SC, the Bab el-Mandeb Strait, the Hormuz Strait, the Malacca Strait, and the Panama Canal, by calculating the chokepoint rate from Lloyd's vessel movement database. Wu et al. (2019) [15] used the LS data of 100 major MCS companies to estimate the impact on the network if the three major chokepoints, including the Malacca Strait, the SC, and the Panama Canal, were blocked and it was discovered that the transport capacity of the entire network could be reduced by 10% to 50%. Lhomme (2015) [16] assessed the global vulnerability of world maritime shipping and identified the most critical ports by evaluating the importance of vertices or edges in a graph. Viljoen and Joubert (2016) [17] simulated the impact of large-scale service reconfiguration affecting priority links by evaluating link-based disruption strategies on a global MCS network constructed from AIS data. They found that the network is by and large robust to such reconfiguration and that some specific strategies for cutting links could decrease the efficiency of the network at the same time. These model simulations are useful for predicting potential failures of maritime networks in the future, whereas the accumulation of empirical knowledge is essential for verifying the validity of the models.

Furthermore, some studies conducted an empirical analysis of past changes. Ducruet et al. (2015) [18] used Lloyd's List records on vessel movements from 1890 to 2008 and analyzed the change in the structure of the global maritime shipping network. Similarly, Ducruet (2017) [19] used Lloyd's data on the inter-port movement of vessels from 1977 to 2008 to calculate the hierarchical structure of ports. Tsubota et al. (2017) [20] used the same data and focused on South Asia from the perspective of the end of the British Empire. They revealed the impact of independence in each country on intra-regional shipping between ports in these countries. The results of these studies provide a significant amount of information on the long-term transition of the entire network structure. However, as the analyses were conducted without distinguishing vessel types, the information obtained on containerships was limited. Ducruet et al. (2016) [21] applied a network analysis on maritime flows connecting cities of the world over the period of 1950–1990 using shipping movement data from Geopolis and Lloyd's databases. From this study, they suggested that the largest cities have maintained their dominance in terms of network centrality and geographical reach. However, this study did not focus on MCS. Ducruet et al. (2019) [22] used fully cellular MCS data from 1977 to 2016 to investigate the evolution of inter- and intra-port vessel movements. In this analysis, they measured the average time that ships stay at or voyage between ports and demonstrated the acceleration of global shipping both within and between ports for 40 years. They indicated that larger ports performed better than smaller ones in terms of staying time and that navigating speeds in the longest and shortest shipping links were faster than others.

Meanwhile, some studies have focused on the impact of specific past events. Rousset and Ducruet (2020) [23] analyzed the impacts of historical events that caused port shutdowns, such as the Hanshin-Awaji Earthquake, the September 11 terrorist attacks on the United States, and Hurricane Katrina. As a result, it was confirmed that containerships were relatively sensitive to changes at import/export and transshipment ports because of such events, whereas the impact on the global MCS network was unlikely to spread because the regional MCS network absorbed the impact. Grenzeback and Lukmann (2008) [24] examined the transport sector's response to and recovery from Hurricanes Katrina and Rita and reviewed the influence of the disaster on the national-level movement of freight.

Xu and Itoh (2018) [25] focused on the Hanshin-Awaji Earthquake and analyzed its impact on container cargo flow. They found that, although the ports of Tokyo and Yokohama were not directly affected by the earthquake, extensive diversions of container traffic to the port of Busan occurred from these ports, not only from the port of Kobe.

The 1970s is known as the era of two oil shocks and the reopening of the SC in the maritime shipping market. These events affected the cost of fuel and shipping distance. Some studies on these events have focused on their economic effects from the perspective of historical economics. Feyrer (2021) [26] estimated the effect of distance on trade and the effect of trade on income in the events of the closure of the SC in 1967 and its reopening in 1975 by using IMF trade data. He suggested that the elasticity of trade with respect to the shock is larger when estimated on closure compared with reopening. Parinduri (2012) [27] investigated the relationship between trade and economic growth by observing the impact of the closure of the SC using the gravity model and found that trade led to higher economic growth rather than trade trends before and after the closure. However, these studies did not analyze the structural changes in MCS. Even though the influences of the reopening of the SC on distance and income were smaller than those of the closure, the network structure of MCS can be greatly changed for the reopening of the SC.

Based on the above discussion, we conduct an empirical analysis of the emergence of global MCS using LS data around the 1970s. In particular, we focus on the impacts of the reopening of the SC in 1975, which is thought to have had a significant impact on the MCS network. In the analyses, we use two kinds of LS data, which were preliminarily compiled as the LS network, as will be explained in Section 3, to focus more on the LS network structure, including both GDL and GAL networks. This study aims to accumulate useful knowledge for the methodology of vulnerability analysis and empirical results on MCS in the 1970s, suggesting future impacts on the global MCS network of partial failures in the network.

## 3. Data, Methods, and Models
### 3.1. Data

This study first prepares the global LS network data at 30 time points. Among them, the LS network data at 28 points from February 1969 to December 1995 are acquired from the NYK report, which was compiled based on Lloyd's data, and those in the two other time points (2003 and 2016) are from the MDS containership databank. The data include information on the LS of both full-container and semi-container ships, such as the operating companies, ports of call, their orders, frequencies, names of containerships, deadweight tonnages of the vessels, and maximum loadable number of containers in twenty-foot equivalent units (TEU), as shown in Figure 1. In contrast to pure vessel movement data, such as Lloyd's, these databases were preliminarily compiled on an LS basis and thus enable more precise analyses of the LS network.

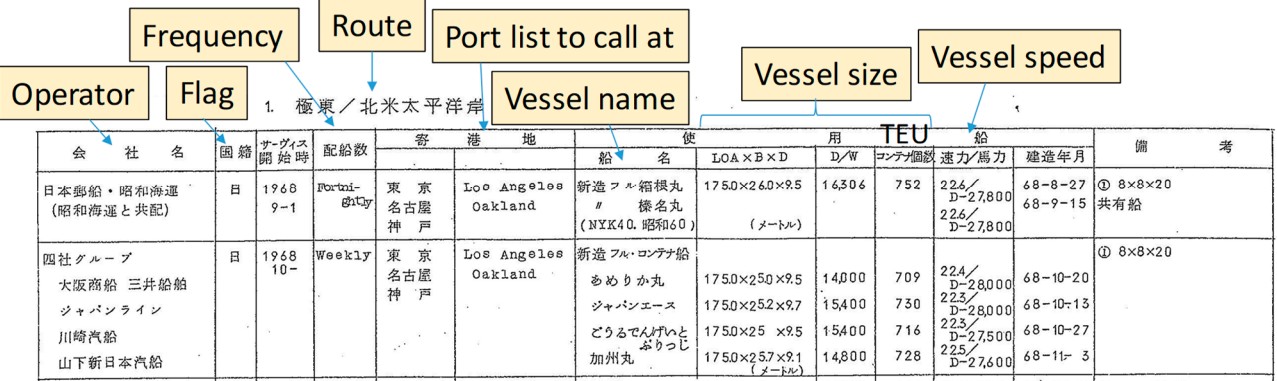

**Figure 1.** Sample of NYK report (issued in February 1969).

As the NYK reports were published in paper format, they should first be digitized in the same format as the MDS data. First, we exclude the data points for which volumes are obviously lower than those of nearby years; consequently, 16 data points are selected to be as equal as possible in terms of time interval. Subsequently, errors are eliminated from the data at each data point selected in the previous step. It is notable that, as clearly stated in the NYK reports, the data from 1981 to 1995 did not include regional routes, such as intra-Asian and intra-European routes. Specifically, as shown in Figure 2, the number of ports included in the data as of 1986 and later is un-naturally small. Similarly, the number in 1979 is smaller than in the years before and after. Therefore, considering that the objective of the study focuses on the analysis in the 1970s, the dataset of eight time points, namely, February 1969 (which represents the first half of the year), January 1971, 1973 (which represents the whole year), 1975, 1976, 1981, 2003, and 2016, is finally selected for the analysis. Note that although the data as of 1981 are included in the following analyses, they cannot be used for intra-regional analysis.

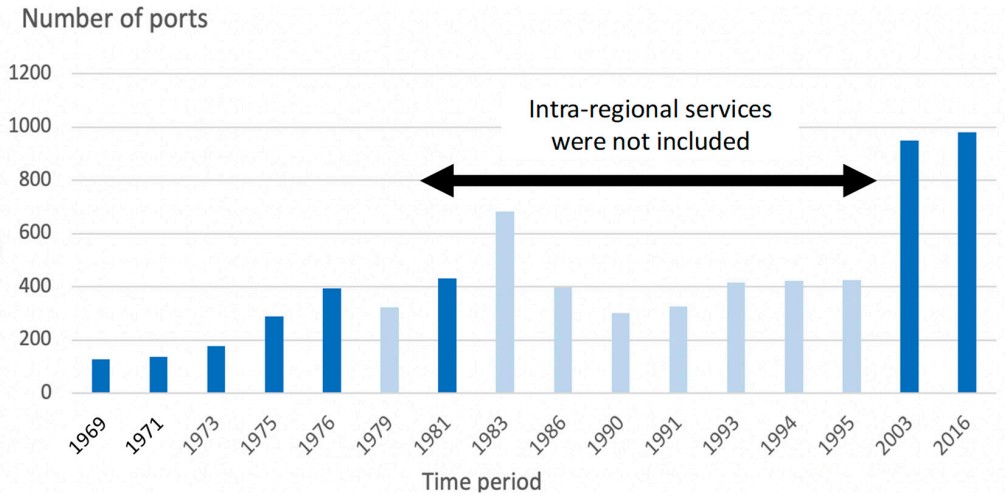

**Figure 2.** Number of container ports included in the data at each selected time point.

*3.2. Methods*

We apply graph theory to the above LS network data; namely, degree, density, and betweenness centrality are used as the indices in the network analysis.

*Degree* is the number of edges extending from one node, whereas *density* is the probability of the existence of edges between nodes in a network, defined as

$$D = \frac{m}{n(n-1)} \tag{1}$$

where $D$ is the density, $n$ is the number of nodes, and $m$ is the number of edges.

*Centrality* is an index that measures the importance of each node in a network. In this study, betweenness centrality, $C_B(v)$, representing whether each node is located on the shortest path of the pairs of other nodes as defined in Equation (2), is used, considering the characteristics of the global MCS to form the hub-and-spoke structure:

$$C_B(v) = \sum_{s \neq v \in V} \sum_{\neq vt \in V} \frac{\sigma_{st}(v)}{\sigma_{st}} \tag{2}$$

where $V$ is a set of nodes, $\sigma_{st}$ is the total number of routes from node $s$ to node $t$, and $\sigma_{st}(v)$ is the number of routes from node $s$ to node $t$ via node $v$. For the clustering method, modularity-based community detection [28], $Q$, defined in Equation (3), is adopted:

$$Q = \frac{1}{2m} \sum_{i \in V} \sum_{j \in V} \left( A_{ij} - \frac{k_i k_k}{2m} \right) \delta(c_i, c_j) \tag{3}$$

where $A_{ij}$ is the $(i, j)$ component in the network adjacency matrix, $k_i$ is the degree of node $i$, $m$ is the number of links ($m = \frac{1}{2} \sum_{i \in V} k_i$), $c_i$ is the community to which node $i$ belongs, and $\delta(c_i, c_j)$ is the Kronecker delta, which is 1 if $c_i = c_j$ and 0 otherwise. Equation (3) indicates that the community of each node is determined such that the density within each community ($\left( A_{ij} - \frac{k_i k_j}{2m} \right) \cdot \delta(c_i, c_j)$) is higher than that of the entire network, as well as the densities between different communities. To save calculation time, this study adopted the Louvain method [29], an iterative calculation algorithm of modularity optimization and community aggregation, to estimate communities, which is adopted as standard in NetworkX, a Python library.

*3.3. Models*

To analyze the network from multiple viewpoints, we develop six different models by changing the method of edge construction and the definition of the weight of edges in the network, as summarized in Table 1. There are two methods of edge construction: a graph of direct linkages (GDL) and a graph of all linkages (GAL). The GDL uses the movement of the containerships directly as edges. In contrast, the GAL connects all ports that are called at in the same LS, with edges. The GDL is suitable for the analysis of the movement network of containerships, such as the first and last ports of call in the region, whereas the GAL is suitable for the analysis of hub-and-spoke structures with transshipments because it can differentiate the ports that are connected in the same LS and the ports that are only connected indirectly through the transshipment port(s).

**Table 1.** Models used in this study with settings on edges.

| Model Names | Edge Linkage | Edge Weight |
|:---:|:---:|:---:|
| GDL-1 | GDL | None |
| GDL-2 | GDL | Shipping Frequency |
| GDL-3 | GDL | Shipping Capacity |
| GAL-1 | GAL | None |
| GAL-2 | GAL | Shipping Frequency |
| GAL-3 | GAL | Shipping Capacity |

GDL: a graph of direct linkages; GAL: a graph of all linkages.

The weights of the edges are defined in three different ways: no weight, shipping frequency, and shipping capacity. The non-weight model, in which the edges are all treated equivalently, is suitable for analyzing the geometry of the networks. The shipping frequency model represents the annual number of vessel movements between ports. The shipping capacity model represents the annual vessel capacities (which are acquired by multiplying annual frequency by average capacity per vessel) between ports. These weighted models can express the difference in the intentions of the inter-port connections based on the actual shipping situation.

Gephi visualization software is used, and Force Atlas is selected as the arrangement algorithm. The size of each node represents its centrality, and the color represents the community to which each node belongs. Figure 3 shows the visualization of the network developed by the GDL-1 and GAL-1 models for 2016, which is the latest year among the available data. In the GDL-1 model, the first and last ports of call in the region, such as Antwerp (Belgium) and Singapore, tend to gain more centralities as expected. Meanwhile, it seems that the GAL-1 model can focus more on the hub ports; however, it is not suitable for the analysis of the shape of the networks through visualization because the density of the network is too high. Hereafter, GDL models are used for the analysis of the movement network of containerships, whereas GAL models are used when focusing on LS networks and container transshipment.

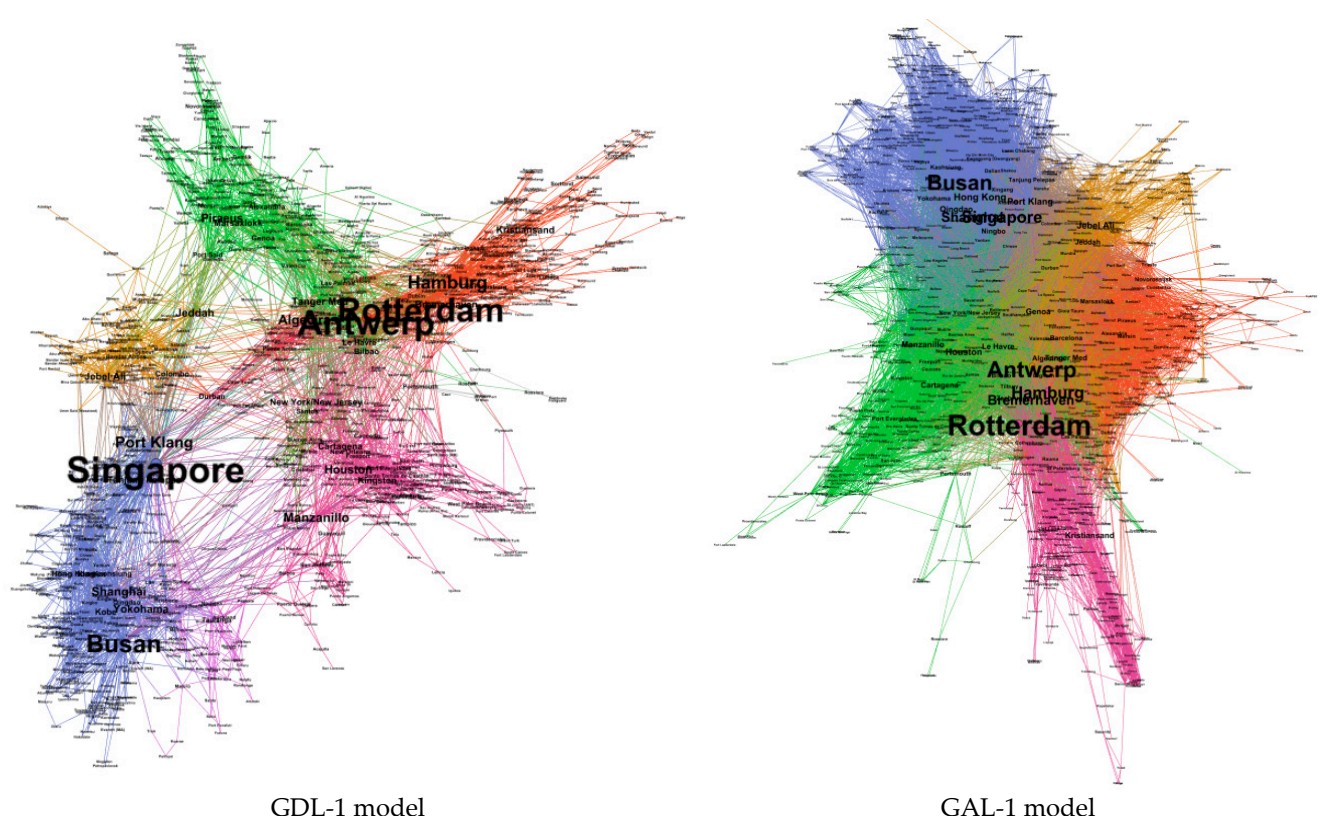

GDL-1 model                                    GAL-1 model

**Figure 3.** MCS network developed by the GDL-1 and GAL-1 models (2016).

## 4. Overview of the Global MCS Network Development (1969–2016)

This section gives an overview of the development history of the MCS network from 1969 to 2016, before focusing on it in the 1970s because understanding the entire history of the MCS network development from its beginning to the present is necessary for positioning the MCS network in the 1970s. Figure 4 tabulates the top 20 ports in the betweenness centrality with their scores estimated from the GDL-1 and GAL-1 models for each analysis year. The figure also marks three major regions: North America (NA), Europe, and EA. Figure 5 summarizes the number of nodes and edges and the network density in the GDL-1 model for each analysis year. The figure reveals that the number of nodes and edges monotonically increases, and the network density decreases from a long-term perspective.

The following analyses in this section focus on the MCS network at four time points, namely, 1969, 1981, 2003, and 2016, to give an overview of their development history. Figure 6 represents the MCS network developed by the GDL-1 model as of 1969, 1981, and 2003. The networks shown in Figures 3 and 6 indicate that the network centered on NA ports represented by New York (NY) in 1969 had changed to a multipolar structure with multiple regions and ports in 1981, then formed a hub-and-spoke structure in and after 2003, in which small ports were clustered around regional hub ports connected by a relatively small number of edges. Comparing the top 20 ports in the betweenness centrality estimated from the GDL-1 model as of 1981, which is shown in Figure 4, with those as of 1969, European ports became more central, and the number of EA ports in the top 20 ports increased. These findings also support our interpretation that the network, as of 1981, had become a multipolar structure. Furthermore, the betweenness centralities of EA ports were comparable to those of European ports in 2003, and the betweenness centralities of ports in other regions increased in 2016.

## GDL-1 model

| # | 1969 | | 1971 | | 1973 | | 1975 | | 1976 | | 1981 | | 2003 | | 2016 | |
|---|---|---|---|---|---|---|---|---|---|---|---|---|---|---|---|---|
| 1 | New York | 0.541 | New York | 0.318 | New York | 0.343 | New York | 0.251 | New York | 0.217 | Hamburg | 0.115 | Singapore | 0.189 | Singapore | 0.148 |
| 2 | Los Angeles | 0.159 | Kobe | 0.163 | Rotterdam | 0.124 | New Orleans | 0.095 | Rotterdam | 0.115 | Rotterdam | 0.101 | Antwerp | 0.143 | Antwerp | 0.137 |
| 3 | Callao | 0.114 | Rotterdam | 0.119 | Kobe | 0.123 | Baltimore | 0.082 | Le Havre | 0.082 | New York | 0.086 | Rotterdam | 0.112 | Rotterdam | 0.134 |
| 4 | Yokohama | 0.111 | Los Angeles | 0.117 | Los Angeles | 0.073 | Hong Kong | 0.074 | New Orleans | 0.071 | Leghorn | 0.077 | Busan | 0.079 | Busan | 0.094 |
| 5 | Liverpool | 0.111 | Hampton Roads | 0.109 | Le Havre | 0.069 | Hamburg | 0.067 | Hamburg | 0.063 | Antwerp | 0.069 | Hamburg | 0.067 | Hamburg | 0.065 |
| 6 | Rotterdam | 0.101 | Baltimore | 0.107 | Hong Kong | 0.061 | Le Havre | 0.060 | Singapore | 0.052 | Los Angeles | 0.064 | Yokohama | 0.044 | Port Klang | 0.053 |
| 7 | Kobe | 0.093 | Balboa | 0.072 | Hampton Roads | 0.060 | London | 0.056 | Liverpool | 0.051 | Houston | 0.063 | Port Klang | 0.042 | Algeciras | 0.043 |
| 8 | Hamburg | 0.080 | Rio de Janeiro | 0.069 | Rio de Janeiro | 0.049 | Charleston | 0.055 | Kobe | 0.045 | Le Havre | 0.062 | Piraeus | 0.041 | Manzanillo | 0.042 |
| 9 | London | 0.071 | Portland | 0.058 | Savannah | 0.049 | Philadelphia | 0.054 | Yokohama | 0.045 | Sydney | 0.060 | Gioia Tauro | 0.040 | Shanghai | 0.038 |
| 10 | Guayaquil | 0.068 | Genoa | 0.057 | Genoa | 0.048 | Antwerp | 0.053 | Savannah | 0.040 | Singapore | 0.058 | Le Havre | 0.040 | Houston | 0.037 |
| 11 | Vancouver | 0.067 | Hamburg | 0.055 | Vancouver | 0.047 | Rotterdam | 0.049 | Los Angeles | 0.040 | Jeddah | 0.057 | Hong Kong | 0.039 | Piraeus | 0.036 |
| 12 | Baltimore | 0.064 | Liverpool | 0.053 | Liverpool | 0.046 | Los Angeles | 0.049 | Antwerp | 0.039 | Yokohama | 0.055 | Manzanillo | 0.031 | Jeddah | 0.033 |
| 13 | Bremerhaven | 0.061 | Sydney | 0.048 | Guayaquil | 0.045 | Savannah | 0.040 | Gothenburg | 0.036 | New Orleans | 0.044 | Bremerhaven | 0.030 | Yokohama | 0.033 |
| 14 | Antwerp | 0.058 | Salvador | 0.048 | Baltimore | 0.043 | Singapore | 0.037 | Vancouver | 0.035 | Montreal | 0.042 | New York | 0.030 | Tanger Med | 0.029 |
| 15 | Kingston | 0.053 | Callao | 0.048 | Leghorn | 0.041 | Houston | 0.034 | Hong Kong | 0.033 | Kobe | 0.040 | Felixstowe | 0.027 | Jebel Ali | 0.029 |
| 16 | Valparaiso | 0.052 | Yokohama | 0.046 | Hamburg | 0.041 | Kobe | 0.032 | Piraeus | 0.031 | Hong Kong | 0.040 | Valencia | 0.025 | Bremerhaven | 0.028 |
| 17 | Buenaventura | 0.047 | Puget Sound | 0.042 | Houston | 0.040 | Bremerhaven | 0.030 | Baltimore | 0.031 | Liverpool | 0.035 | Genoa | 0.025 | Kristiansand | 0.027 |
| 18 | Acajutla | 0.047 | Shimizu | 0.039 | Balboa | 0.036 | Port Klang | 0.028 | Port Klang | 0.030 | Vancouver | 0.029 | Durban | 0.024 | Tanjung Pelepas | 0.025 |
| 19 | Panama | 0.046 | London | 0.037 | New Orleans | 0.035 | Liverpool | 0.027 | London | 0.028 | Mumbai | 0.028 | Bilbao | 0.024 | Cartagena | 0.025 |
| 20 | Bremen | 0.040 | Seattle | 0.036 | Norfolk | 0.034 | Boston | 0.025 | Houston | 0.027 | Bremen | 0.027 | Houston | 0.023 | Kingston | 0.024 |

## GAL-1 model

| # | 1969 | | 1971 | | 1973 | | 1975 | | 1976 | | 1981 | | 2003 | | 2016 | |
|---|---|---|---|---|---|---|---|---|---|---|---|---|---|---|---|---|
| 1 | New York | 0.156 | New York | 0.124 | New York | 0.131 | New York | 0.144 | Rotterdam | 0.080 | Hamburg | 0.081 | Antwerp | 0.137 | Rotterdam | 0.099 |
| 2 | Los Angeles | 0.151 | Los Angeles | 0.113 | Baltimore | 0.090 | Baltimore | 0.118 | Newyork | 0.073 | Rotterdam | 0.078 | Rotterdam | 0.121 | Antwerp | 0.075 |
| 3 | Liverpool | 0.149 | Baltimore | 0.097 | Los Angeles | 0.077 | Philadelphia | 0.066 | New Orleans | 0.063 | Antwerp | 0.067 | Singapore | 0.062 | Busan | 0.071 |
| 4 | Baltimore | 0.120 | Rotterdam | 0.057 | Houston | 0.045 | New Orleans | 0.056 | Baltimore | 0.063 | Leghorn | 0.044 | Hamburg | 0.062 | Hamburg | 0.053 |
| 5 | Rotterdam | 0.110 | Portland | 0.049 | Philadelphia | 0.042 | Antwerp | 0.041 | Antwerp | 0.045 | Liverpool | 0.033 | Busan | 0.054 | Singapore | 0.052 |
| 6 | Antwerp | 0.109 | Philadelphia | 0.045 | Benghazi | 0.040 | Hong Kong | 0.040 | Houston | 0.045 | Newyork | 0.032 | Hong Kong | 0.041 | Shanghai | 0.042 |
| 7 | Portland | 0.087 | Seattle | 0.040 | Vancouver | 0.036 | Savannah | 0.036 | Le Havre | 0.044 | Houston | 0.032 | Bremerhaven | 0.028 | Bremerhaven | 0.041 |
| 8 | Yokohama | 0.074 | Antwerp | 0.039 | Rotterdam | 0.036 | Charleston | 0.035 | Hamburg | 0.044 | Hong Kong | 0.029 | Piraeus | 0.023 | Hong Kong | 0.026 |
| 9 | Philadelphia | 0.067 | Vancouver | 0.039 | Seattle | 0.035 | Rotterdam | 0.034 | Philadelphia | 0.036 | Yokohama | 0.028 | Port Klang | 0.022 | Jebel Ali | 0.024 |
| 10 | Hamburg | 0.049 | Liverpool | 0.036 | Norfolk | 0.030 | Houston | 0.033 | Vancouver | 0.026 | Le Havre | 0.028 | Felixstowe | 0.022 | Port Klang | 0.024 |
| 11 | Norfolk | 0.047 | Kobe | 0.032 | New Orleans | 0.028 | Los Angeles | 0.028 | Liverpool | 0.025 | Singapore | 0.027 | Le Havre | 0.021 | Houston | 0.023 |
| 12 | Montreal | 0.046 | Hamburg | 0.030 | Le Havre | 0.029 | Hamburg | 0.027 | Norfolk | 0.025 | Kobe | 0.026 | Seattle | 0.021 | Manzanillo | 0.021 |
| 13 | Sanfrancisco | 0.041 | London | 0.029 | Portland | 0.028 | Le Havre | 0.024 | Kobe | 0.024 | Genoa | 0.025 | Yokohama | 0.020 | Qingdao | 0.020 |
| 14 | Nagoya | 0.036 | Yokohama | 0.027 | Liverpool | 0.025 | Benghazi | 0.021 | Los Angeles | 0.022 | Sydney | 0.024 | Barcelona | 0.020 | Kristiansand | 0.020 |
| 15 | London | 0.031 | Balboa | 0.026 | London | 0.024 | Seattle | 0.020 | Leghorn | 0.020 | Baltimore | 0.024 | Genoa | 0.020 | Genoa | 0.020 |
| 16 | Hampton Roads | 0.031 | Sydney | 0.022 | Hong Kong | 0.023 | Genoa | 0.018 | Charleston | 0.020 | New Orleans | 0.022 | Trieste | 0.018 | Le Havre | 0.018 |
| 17 | Guayaquil | 0.029 | Melbourne | 0.022 | Kobe | 0.022 | Norfolk | 0.018 | Savannah | 0.020 | Oakland | 0.020 | Kaohsiung | 0.017 | Cartagena | 0.018 |
| 18 | Buenaventura | 0.029 | San Francisco | 0.022 | Hamburg | 0.021 | Singapore | 0.017 | Marseilles | 0.020 | Melbourne | 0.020 | Shanghai | 0.016 | Tanger Med | 0.018 |
| 19 | Callao | 0.024 | Hampton Roads | 0.021 | Yokohama | 0.021 | Portland | 0.016 | Singapore | 0.019 | San Francisco | 0.018 | Houston | 0.016 | Jeddah | 0.017 |
| 20 | Kobe | 0.024 | Nagoya | 0.018 | Nagoya | 0.016 | Bremerhaven | 0.016 | Bremen | 0.018 | Vancouver | 0.018 | Genoa | 0.015 | Algeciras | 0.017 |

Legend: North America | Europe | East Asia

**Figure 4.** Top 20 ports in betweenness centrality and their scores estimated from the GDL-1 and GAL-1 models (1969–2016).

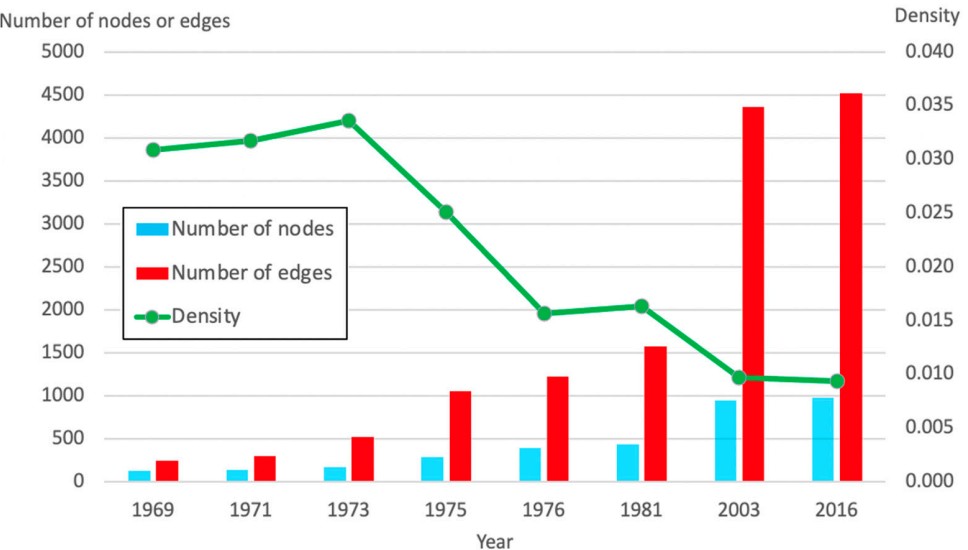

**Figure 5.** Number of nodes and edges and network density in the GDL-1 model (1969–2016).

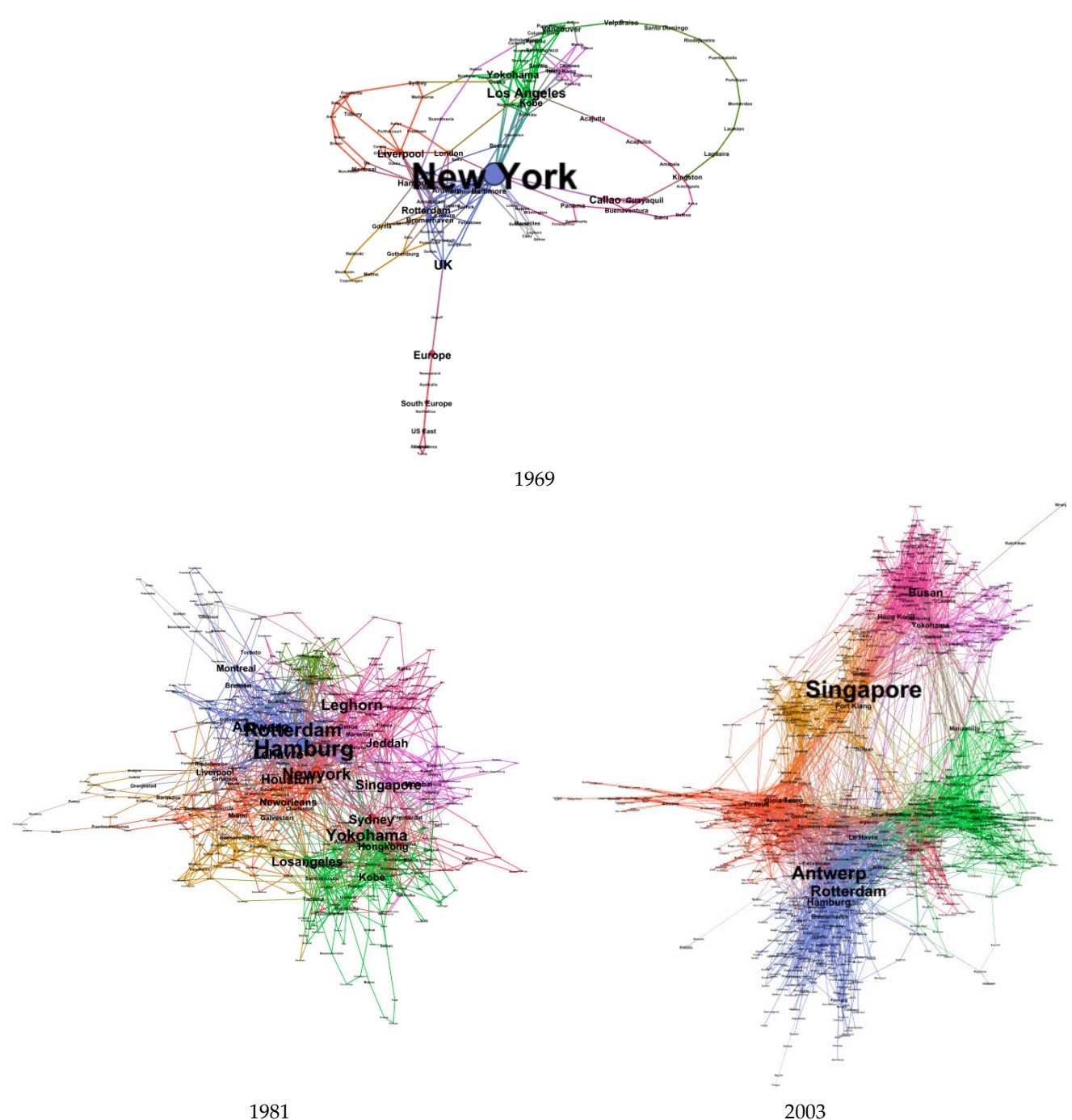

**Figure 6.** MCS network developed by the GDL-1 model in each year.

The top 20 ports in the betweenness centrality estimated from the GAL-1 model were slightly different from those from the GDL-1 model, as shown in Figure 4; namely, the betweenness centralities of the ports in a specific region were dominant in the GAL-1 model, such as NA in the early 1970s. A possible reason for the difference is that, in the GAL-1 model, almost all ports in high centrality areas were connected to each other and on the shortest path. For some ports, such as Singapore, Port Klang (Malaysia), and Yokohama (Japan), the rank in the GAL-1 model was lower than that in the GDL-1 model, whereas the rank was higher for some ports, such as Rotterdam (the Netherlands), in the GAL-1 model than in the GDL-1 model. This is because the centrality of the last or first ports in the region tends to be larger in the GDL-1 model. In contrast, in hub ports other than the

last or first ports in the region, the rank in the GAL-1 model tends to be higher than that in the GDL-1 model.

Table 2 summarizes the number of major communities in the networks of the GDL-1, 2, and 3 models for each time point. Note that the 'major' community is defined as the one that occupies 2% or more of the total number of ports at each time point, considering that the total number of ports at each time point ranges between 100 and 1000. The table indicates that the number of major communities gradually increased until 1981 in a similar manner in each model, but there was a gap among the three models after 2003; the numbers in the GDL-2 model were much larger than those in the GDL-1 and GDL-3 models in 2003 and 2016. This result implies that the GDL-2 model can distinguish between ports connected by trunk routes of global MCS provided by larger vessels with a relatively lower frequency and those connected by feeder transports provided by smaller vessels with relatively higher frequency. For example, the number of communities was often larger in the GDL-2 model in the world major regions, such as the North Sea, Mediterranean Sea, Pacific Ocean, and Indian Ocean, where the trunk route and feeder transport services were mixed.

**Table 2.** Number of major communities in the GDL-1, 2, and 3 model networks.

|  | 1969 | 1971 | 1973 | 1975 | 1976 | 1981 | 2003 | 2016 |
|---|---|---|---|---|---|---|---|---|
| **GDL-1** | 9 | 9 | 8 | 9 | 9 | 10 | 7 | 8 |
| **GDL-2** | 8 | 9 | 9 | 10 | 12 | 12 | 12 | 12 |
| **GDL-3** | 7 | 7 | 8 | 10 | 11 | 11 | 8 | 9 |

Figure 7 tabulates the top 20 ports in the betweenness centrality with their scores estimated from the GDL-2 and GDL-3 models for each analysis year. The figure implies that the GDL-3 model tends to give a larger betweenness centrality not only for the first or last ports of the regions that feeder ports are less connected with, such as Japanese ports, but also for the tentative attractive ports due to war, such as Saigon (Vietnam) in 1973 and Umm Qasr (Iraq) in 2003. Moreover, we find that compared with these results on the GDL-2 and GDL-3 models, in the GDL-1 model (whose results are shown in Figure 4), regional hub ports that many feeder ports are directly connected with tend to earn larger scores, such as Rotterdam, Hamburg (Germany), Busan (South Korea), and Singapore.

**GDL-2 model**

| # | 1969 | | 1971 | | 1973 | | 1975 | | 1976 | | 1981 | | 2003 | | 2016 | |
|---|---|---|---|---|---|---|---|---|---|---|---|---|---|---|---|---|
| 1 | New York | 0.424 | New York | 0.339 | New York | 0.300 | New York | 0.181 | Singapore | 0.137 | Rotterdam | 0.221 | Antwerp | 0.174 | Antwerp | 0.246 |
| 2 | Los Angeles | 0.130 | Kobe | 0.152 | Rotterdam | 0.232 | Antwerp | 0.103 | New York | 0.132 | Apapa | 0.162 | Singapore | 0.139 | Singapore | 0.208 |
| 3 | Yokohama | 0.105 | Shimizu | 0.116 | Baltimore | 0.106 | Savannah | 0.089 | New Orleans | 0.104 | Leghorn | 0.150 | Aqaba | 0.096 | Houston | 0.123 |
| 4 | Callao | 0.097 | Liverpool | 0.112 | Tokyo | 0.104 | Singapore | 0.071 | Baltimore | 0.088 | Singapore | 0.126 | Rotterdam | 0.095 | Hamburg | 0.113 |
| 5 | Kobe | 0.092 | Vancouver | 0.108 | Hong Kong | 0.094 | Amsterdam | 0.069 | Port Klang | 0.084 | Cartagena | 0.115 | Houston | 0.081 | Jeddah | 0.109 |
| 6 | Rotterdam | 0.078 | Baltimore | 0.081 | Kobe | 0.083 | New Orleans | 0.065 | Hamburg | 0.082 | Cadiz | 0.098 | Alexandria | 0.081 | Yokohama | 0.081 |
| 7 | Liverpool | 0.072 | Balboa | 0.080 | Vancouver | 0.065 | London | 0.062 | Los Angeles | 0.065 | Lagos | 0.092 | Fortaleza | 0.074 | Bilbao | 0.068 |
| 8 | Hamburg | 0.069 | Rotterdam | 0.073 | Guayaquil | 0.060 | Felixstowe | 0.054 | Kobe | 0.063 | Sydney | 0.087 | Umm Qasr | 0.072 | Bremerhaven | 0.067 |
| 9 | London | 0.060 | Hampton Roads | 0.067 | Houston | 0.055 | Baltimore | 0.053 | Vancouver | 0.059 | Antwerp | 0.078 | Busan | 0.071 | Rotterdam | 0.064 |
| 10 | Vancouver | 0.059 | Portland | 0.060 | Liverpool | 0.050 | Copenhagen | 0.052 | Brisbane | 0.054 | New York | 0.077 | Kobe | 0.060 | Alexandria | 0.063 |
| 11 | Guayaquil | 0.059 | Rio de Janeiro | 0.056 | Hampton Roads | 0.046 | Vancouver | 0.048 | Dunkirk | 0.053 | Le Havre | 0.073 | Bilbao | 0.060 | Busan | 0.060 |
| 12 | Bremerhaven | 0.058 | Tokyo | 0.056 | Gothenburg | 0.041 | Philadelphia | 0.048 | Barcelona | 0.050 | Hamburg | 0.072 | Karachi | 0.050 | New Orleans | 0.059 |
| 13 | Baltimore | 0.050 | Callao | 0.054 | Rio de Janeiro | 0.039 | Manchester | 0.046 | Liverpool | 0.046 | Houston | 0.071 | Hamburg | 0.043 | Altamira | 0.049 |
| 14 | Kingston | 0.046 | Montreal | 0.051 | Montreal | 0.038 | Galveston | 0.044 | Callao | 0.043 | Piraeus | 0.070 | Manaus | 0.042 | Kobe | 0.046 |
| 15 | Valparaiso | 0.046 | Los Angeles | 0.044 | Savannah | 0.035 | Fremantle | 0.041 | Wellington | 0.043 | Montreal | 0.058 | Lae | 0.041 | Genoa | 0.045 |
| 16 | Buenaventura | 0.041 | San Francisco | 0.041 | Le Havre | 0.035 | Norfolk | 0.040 | Rotterdam | 0.043 | Baltimore | 0.057 | Jeddah | 0.039 | Kristiansand | 0.043 |
| 17 | Acajutla | 0.041 | Yokohama | 0.040 | Galveston | 0.030 | Charleston | 0.039 | Le Havre | 0.040 | Bilbao | 0.056 | Leghorn | 0.035 | Durban | 0.042 |
| 18 | Panama | 0.040 | Seattle | 0.040 | Balboa | 0.030 | Gothenburg | 0.038 | Savannah | 0.038 | Honolulu | 0.053 | Port Klang | 0.035 | Baltimore | 0.041 |
| 19 | Antwerp | 0.039 | Osaka | 0.039 | Manila | 0.029 | Conakry | 0.037 | Yokohama | 0.037 | Los Angeles | 0.051 | Casablanca | 0.033 | Immingham | 0.039 |
| 20 | Laguaira | 0.034 | Salvador | 0.038 | Genoa | 0.027 | Rotterdam | 0.037 | Portland | 0.037 | Vancouver | 0.051 | Hodeidah | 0.032 | Shanghai | 0.032 |

**GDL-3 model**

| # | 1969 | | 1971 | | 1973 | | 1975 | | 1976 | | 1981 | | 2003 | | 2016 | |
|---|---|---|---|---|---|---|---|---|---|---|---|---|---|---|---|---|
| 1 | New York | 0.394 | Kobe | 0.179 | New York | 0.342 | New York | 0.174 | Yokohama | 0.195 | Rotterdam | 0.248 | Umm Qasr | 0.195 | Antwerp | 0.196 |
| 2 | Callao | 0.126 | New York | 0.157 | Rotterdam | 0.164 | Glasgow | 0.095 | New York | 0.180 | Leghorn | 0.190 | Rotterdam | 0.188 | Houston | 0.139 |
| 3 | Los Angeles | 0.118 | Portland | 0.138 | Hong Kong | 0.144 | New Orleans | 0.092 | Rotterdam | 0.154 | Cartagena | 0.154 | Kobe | 0.186 | Yokohama | 0.133 |
| 4 | Yokohama | 0.104 | Shimizu | 0.121 | Baltimore | 0.135 | Copenhagen | 0.090 | Port Klang | 0.147 | Bilbao | 0.153 | Yokohama | 0.158 | Singapore | 0.128 |
| 5 | Kobe | 0.095 | Baltimore | 0.108 | Guayaquil | 0.098 | Baltimore | 0.082 | Singapore | 0.120 | Buenaventura | 0.133 | Antwerp | 0.152 | Alexandria | 0.125 |
| 6 | Rotterdam | 0.074 | Moji | 0.094 | Tokyo | 0.092 | Hong Kong | 0.079 | Nakhodka | 0.108 | Antwerp | 0.131 | Vladivostok | 0.141 | Benghazi | 0.120 |
| 7 | Baltimore | 0.064 | Tacoma | 0.069 | Houston | 0.088 | Philadelphia | 0.074 | Kobe | 0.103 | Lagos | 0.115 | Singapore | 0.103 | Rotterdam | 0.115 |
| 8 | Bremerhaven | 0.060 | Houston | 0.066 | Vancouver | 0.077 | Antwerp | 0.068 | Nagoya | 0.102 | Cadiz | 0.100 | Trondheim | 0.100 | Hamburg | 0.115 |
| 9 | Guayaquil | 0.056 | Seattle | 0.063 | Miami | 0.070 | Gothenburg | 0.063 | Piraeus | 0.098 | Apapa | 0.099 | Aqaba | 0.090 | Bilbao | 0.111 |
| 10 | Vancouver | 0.056 | Puget Sound | 0.061 | Savannah | 0.053 | Houston | 0.053 | Haifa | 0.095 | Singapore | 0.099 | Bilbao | 0.083 | Kobe | 0.108 |
| 11 | Buenaventura | 0.053 | Liverpool | 0.057 | Algiers | 0.051 | Apapa | 0.051 | Portland | 0.087 | Bremerhaven | 0.096 | Genoa | 0.079 | Incheon | 0.102 |
| 12 | Liverpool | 0.045 | Rotterdam | 0.048 | Yokohama | 0.050 | Fremantle | 0.050 | Osaka | 0.087 | Matarani | 0.095 | Houston | 0.075 | Immingham | 0.099 |
| 13 | Kingston | 0.044 | San Francisco | 0.042 | Saigon | 0.047 | Conakry | 0.048 | Oakland | 0.074 | New Orleans | 0.095 | Busan | 0.074 | Kristiansand | 0.089 |
| 14 | Valparaiso | 0.043 | Hamburg | 0.039 | Kobe | 0.047 | London | 0.045 | New Orleans | 0.074 | Sheerness | 0.086 | Tripoli | 0.057 | Genoa | 0.080 |
| 15 | Acajutla | 0.039 | Southampton | 0.031 | Manila | 0.046 | Hamburg | 0.044 | Houston | 0.073 | Houston | 0.082 | | | Fremantle | 0.069 |
| 16 | Osaka | 0.037 | Brisbane | 0.027 | Cadiz | 0.043 | Savannah | 0.043 | Santo Domingo | 0.064 | Galveston | 0.074 | Hodeidah | 0.052 | Osaka | 0.068 |
| 17 | Gothenburg | 0.036 | Callao | 0.022 | Los Angeles | 0.043 | Amsterdam | 0.041 | Liverpool | 0.062 | Los Angeles | 0.073 | Incheon | 0.048 | Flying Fish Cove | 0.068 |
| 18 | Hamburg | 0.035 | Wilmington | 0.022 | Montreal | 0.041 | Newport News | 0.046 | Seattle | 0.050 | Matadi | 0.063 | Alexandria | 0.047 | Busan | 0.066 |
| 19 | Antwerp | 0.034 | Antwerp | 0.020 | Buenaventura | 0.039 | Tokyo | 0.036 | Sheerness | 0.049 | New York | 0.063 | Kingston | 0.039 | Las Palmas | 0.063 |
| 20 | Laguaira | 0.032 | Vancouver | 0.020 | Lisbon | 0.037 | Hampton Roads | 0.036 | Halifax | 0.048 | Hamburg | 0.061 | Jakarta | 0.038 | Casablanca | 0.061 |

North America　　Europe　　East Asia

**Figure 7.** Top 20 ports in betweenness centrality and their scores estimated from the GDL-2 and GDL-3 models (1969–2016).

## 5. Structural Changes in the Emergence of MCS (1969–1981)

We then focus on the period from 1969 to 1981 and changes in the MCS network from a medium-term perspective in the era of the emergence of global MCS. Figure 8 summarizes the number of container ports where at least one LS was connected from 1969 to 1976 for each region of the world, and Figure 9 shows the MCS networks developed by the GDL-1 model in 1971, 1973, 1975, and 1976.

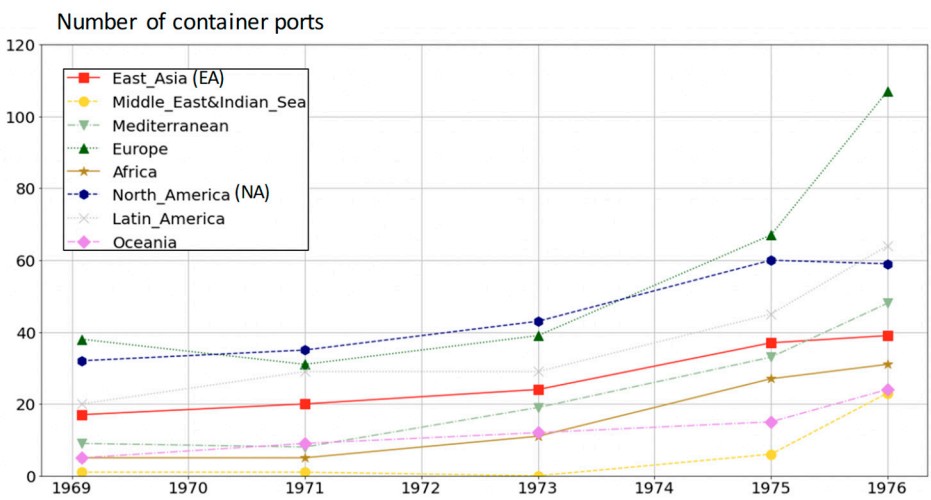

**Figure 8.** Number of container ports in each region of the world (1969–1976).

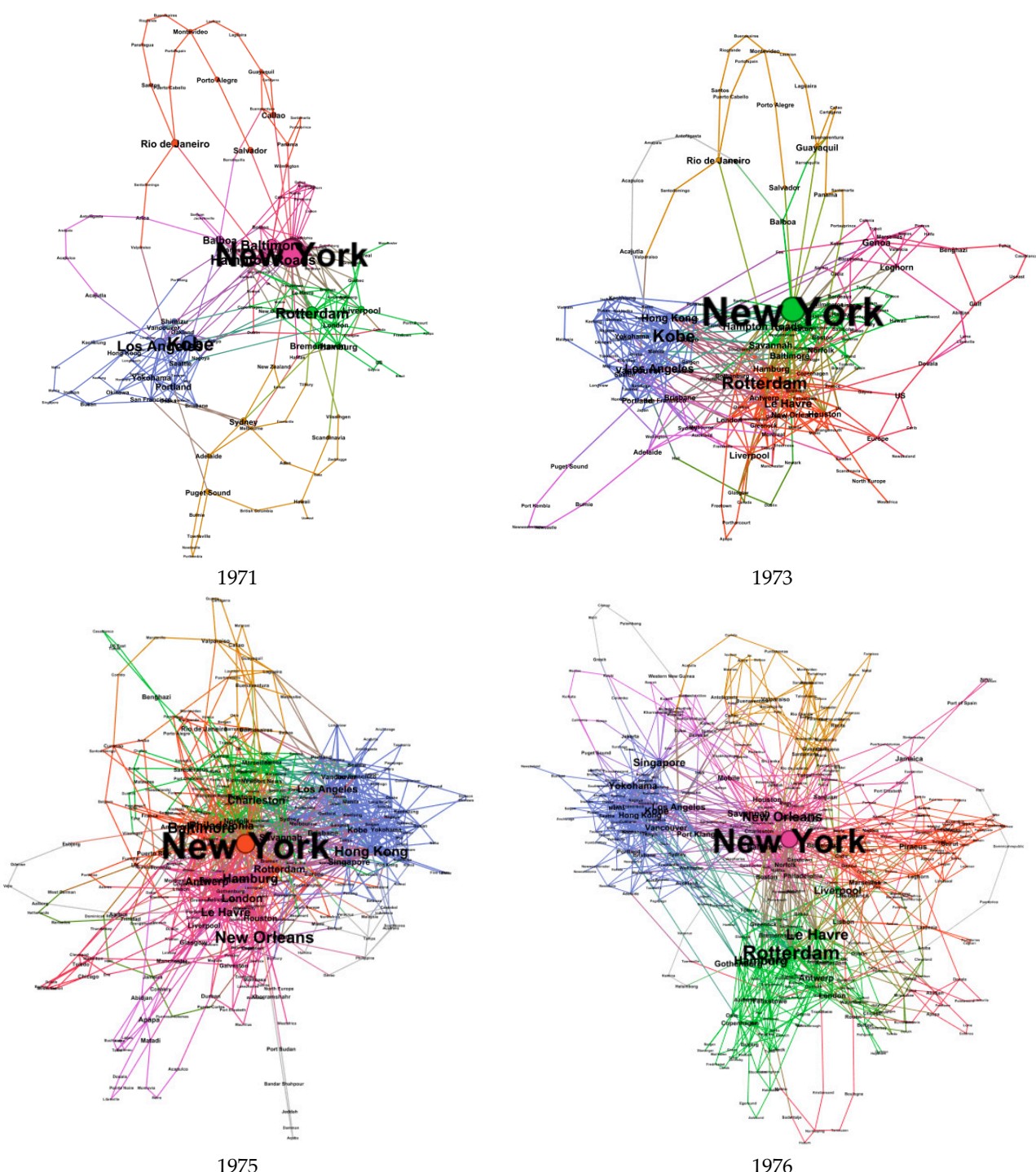

**Figure 9.** MCS network developed by the GDL-1 model in each year of the 1970s.

First, Figure 8 indicates that the number of container ports significantly increased in the middle of 1970s, even in regions outside of the three major regions, such as Latin America and the Middle East. This implies that container shipping rapidly expanded worldwide. In contrast, as indicated in Figure 5 in the previous section, after the network density slightly increased from 1969 to 1973, it dropped until 1976. The port rankings estimated from the GDL-1 model shown in Figure 4 also indicate the trend that the betweenness centrality rankings of NA ports had been high from 1969 to 1975 and then gradually declined since 1976, whereas those of European and EA ports rose. As a result, in 1981, as mentioned in the previous section, the structure of the network changed to multipolar. Figures 6 and 9

indicate that the unipolar network centering on the port of NY had gradually changed to a multipolar structure from 1969 to 1981. Moreover, as shown in Figure 4, European ports took the lead in 1981, and the difference in the average betweenness centrality of major ports between the three major regions was also reduced.

This observation, based on the GDL-1 model, that the NA-centered network structure in 1969 gradually changed to a multipolar structure with three major regions until 1981, is also obtained through the analyses of the network developed by the GAL models. As shown in Figure 4, NA ports were dominant in the betweenness centrality from 1969 to 1975; however, they competed with European ports in 1976 and lost their competitiveness rapidly during and after 1981. Regarding this point, the NYK report (1981) [30] stated that 'the trunk routes of LSs, which connected major regions in the world, were converted from conventional vessels into containerships until the early 1970s, and then the feeder services began to expand from the middle of the 1970s (note that was originally written in Japanese and translated into English by the authors).' Therefore, we conclude that the replacement of conventional LSs with containerships caused diversification of the MCS network in trunk routes, resulting in a reduction in the betweenness centralities of major ports. Moreover, since then, the expansion of the feeder services, which connected new container ports in the developing regions to major ports, enhanced the increase in the number of ports in developing regions.

## 6. Impact of Reopening of the SC in 1975

Following the outbreak of the third Arab–Israeli War in the Middle East in 1967, the SC was blocked for about 8 years. During the blockade, maritime shipping between Europe and Asia was forced to make a long voyage via the Cape of Good Hope at the southernmost point of the African continent. This situation was resolved after minesweeping operations by the U.S. Army in mid-1975 after the fourth Arab–Israeli War (Yom Kippur War). The reopening of the SC greatly reduced the distance of maritime shipping between Europe and EA. In this section, we discuss the impact of the SC reopening on the global MCS network. Note that it is difficult to compare the networks before and after the SC blockade because there was no global MCS network before 1967.

First, Figure 9 indicates that no significant changes were identified in the GDL-1 model networks in 1975 and 1976. We then focus on the differences in the inter- and intra-regional densities. Figure 10 depicts the changes in inter- and intra-regional densities of the three major regions in the GDL-1 model network from 1969 to 1981. Inter-regional density focuses only on the edges connecting two different regions, whereas intra-regional density focuses only on the edges within a region. Note that intra-regional densities in 1981 are not included in the figure because the data after 1981 did not contain information on intra-regional LSs, as described in Section 3.1. As indicated in Figure 10, all the inter-regional densities (including those between Europe and EA) dropped sharply in 1976, whereas all the intra-regional densities did not change significantly from 1975 to 1976.

Subsequently, we apply the GAL model in the following analyses. For the following analyses, we classify each port into several regional groups, as shown in Figure 11, which may share similar impacts of the reopening of the SC related to their location and size based on Shibasaki et al. (2016) [31] and Shibasaki et al. (2017) [32]. Figure 12 summarizes the average number of ports connected to three representative ports in each region's group in the GAL-1 model. As shown in Figure 12, the number of connected ports on both sides of the SC (i.e., group 1: smaller ports in the Mediterranean Sea near the SC; group 4: Red Sea/Indian Ocean ports) increased significantly from 1975 to 1976. This discontinuous change from 1975 to 1976 may have been related to the reopening of the SC.

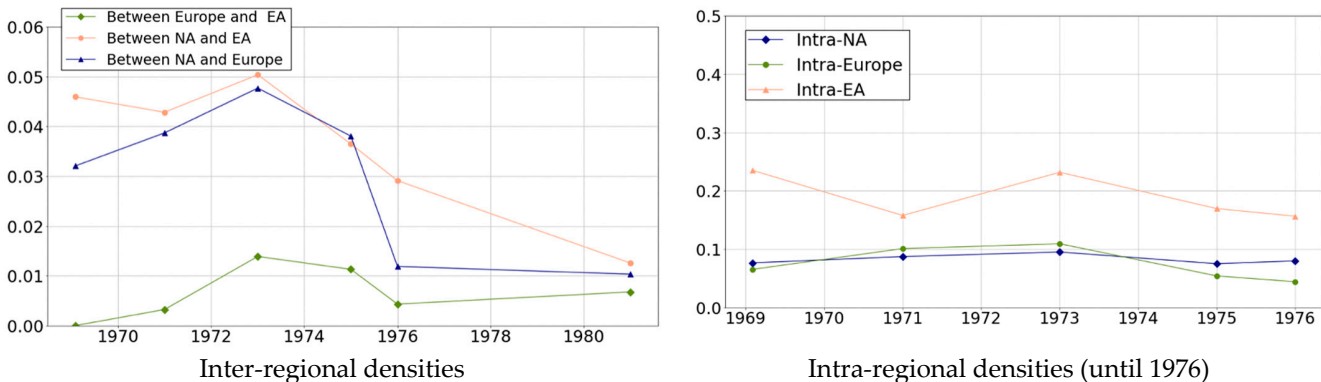

**Figure 10.** Inter-regional and intra-regional densities of major regions in the GDL-1 model network (1969–1981).

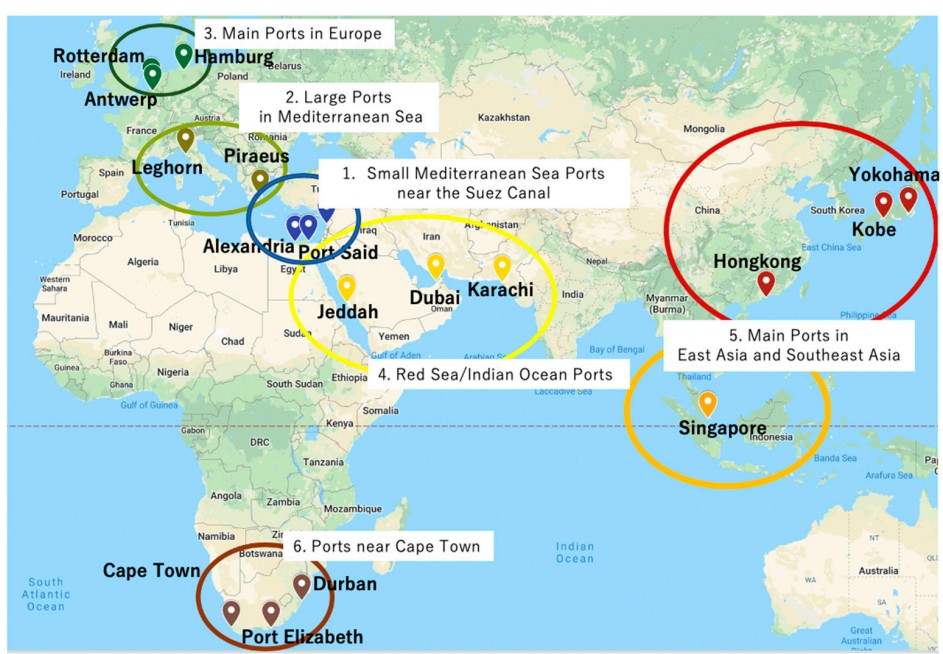

**Figure 11.** Classification of port groups of the world with representative ports.

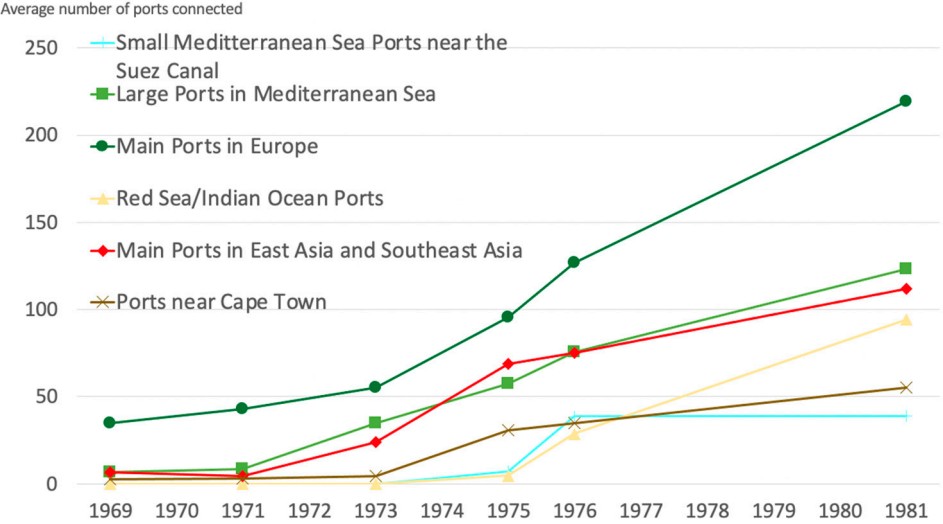

**Figure 12.** Average number of ports connected for each regional group.

For further understanding of the influences of the closure of the SC, we focus on three ports at the sphere of the SC: Jeddah (Saudi Arabia), Dubai (United Arab Emirates), and Alexandria (Egypt). Figure 13 indicates the results in each GAL model in the port of Jeddah as an example of the group 4 port. Although an increase in number was observed in 1976 in the GAL-1 model, it was mainly caused by the increase in intra-regional connections, not connections with the ports at the other end of the SC. Moreover, the results for the GAL-2 and GAL-3 models reveal that the weighted numbers of connected ports in 1975 and 1976 were much smaller, indicating that both the frequency and annual capacity of the LSs called at Jeddah were very low at those times. In contrast, not only did the number of ports connected by the LSs significantly increase in 1981, including connections with European and EA ports, but the frequency of the LSs and the size of the vessels also increased significantly. The same observation was found in the port of Dubai, as indicated in Figure 14. In the GAL-1 model, the port of Dubai was connected to several NA ports via the SC in 1976; however, the weighted numbers of ports connected to them in the GAL-2 and GAL-3 models were very low. Similarly, regarding the ports in group 1, as indicated in Figure 15, as an example of the GAL-1 model in the port of Alexandria, the number of connections increased in 1976; however, it was mostly derived from connections with European ports, not new connections with eastern ports across the SC. Therefore, it cannot be said that the reopening of the SC increased the number of connected ports immediately.

The above analysis does not explain the discontinuous change from 1975 to 1976 in the density of inter-regional connections in Figure 10. Thus, we then summarize the changes in the average number of ports of call per service from 1969 to 1981 in Figure 16 for the inter-regional LSs between the three major regions and the intra-regional LSs in each major region. These numbers are also calculated from the same source acquired from the NYK reports, although the numbers for intra-regional LSs in 1981 are not displayed because of comparability, similar to other figures. The figure indicates that the average number of ports of call per service in all inter-regional LSs sharply increased in 1976 and decreased again in 1981. Among them, the rapid increase in 1976 in the LSs between Europe and EA was caused by an increase in the number of ports of call in Europe and EA, not in the Middle East and South Asia, which are located in the middle of this trunk route, because the container ports in these regions were underdeveloped at that time. The inter-regional density in the GDL-1 model, therefore, declined in 1976 because the increase in the number of ports of call in the inter-regional LSs geographically limited the first and last ports of these regions to some specific ports, such as Antwerp and Singapore.

According to the NYK report (1976) [33], the global MCS market in 1975 faced a surplus of containerships because of the reduction in shipping time caused by the reopening of the SC. Moreover, the NYK report (1976) [33] stated that the number of transported containers in 1975 declined by at least about 20% due to the prolonged recession since 1974 and that LSs operated by Maersk increased the number of ports of call per service. In summary, the increase in the average number of ports of call in the inter-regional LSs (including those between Europe and EA) in 1976 was considered to be a result of the reorganization of LS networks to cope with the decrease in the loading factor of containerships, which was reciprocally caused by the decrease in cargo shipping demand due to the global recession since the first oil shock and the surplus of containerships due to the reopening of the SC. Note that both the first oil shock and the reopening of the SC were consequences of the same event, the Arab–Israeli War.

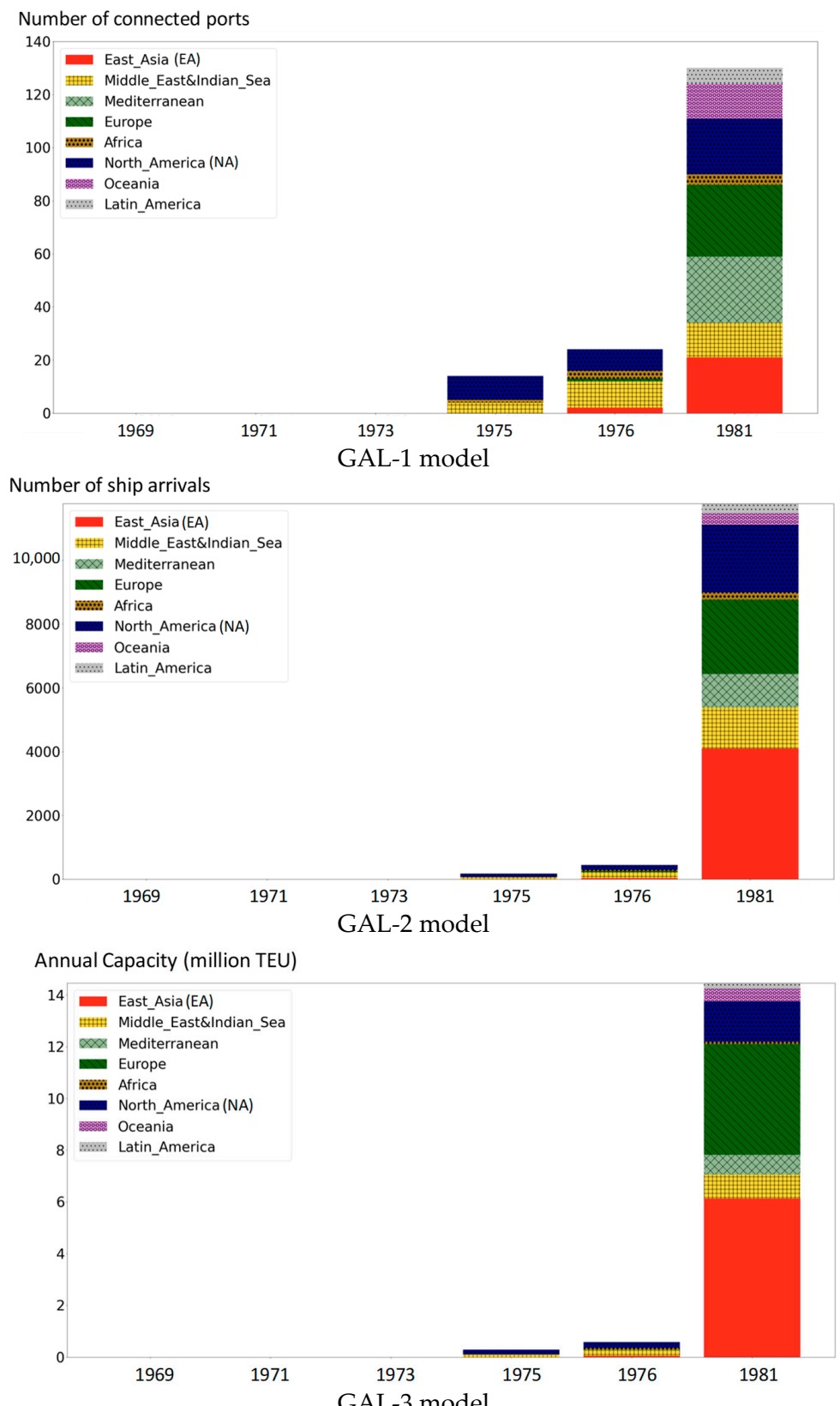

**Figure 13.** (Weighted) number of connected ports and regional breakdown in the port of Jeddah (1969–1981).

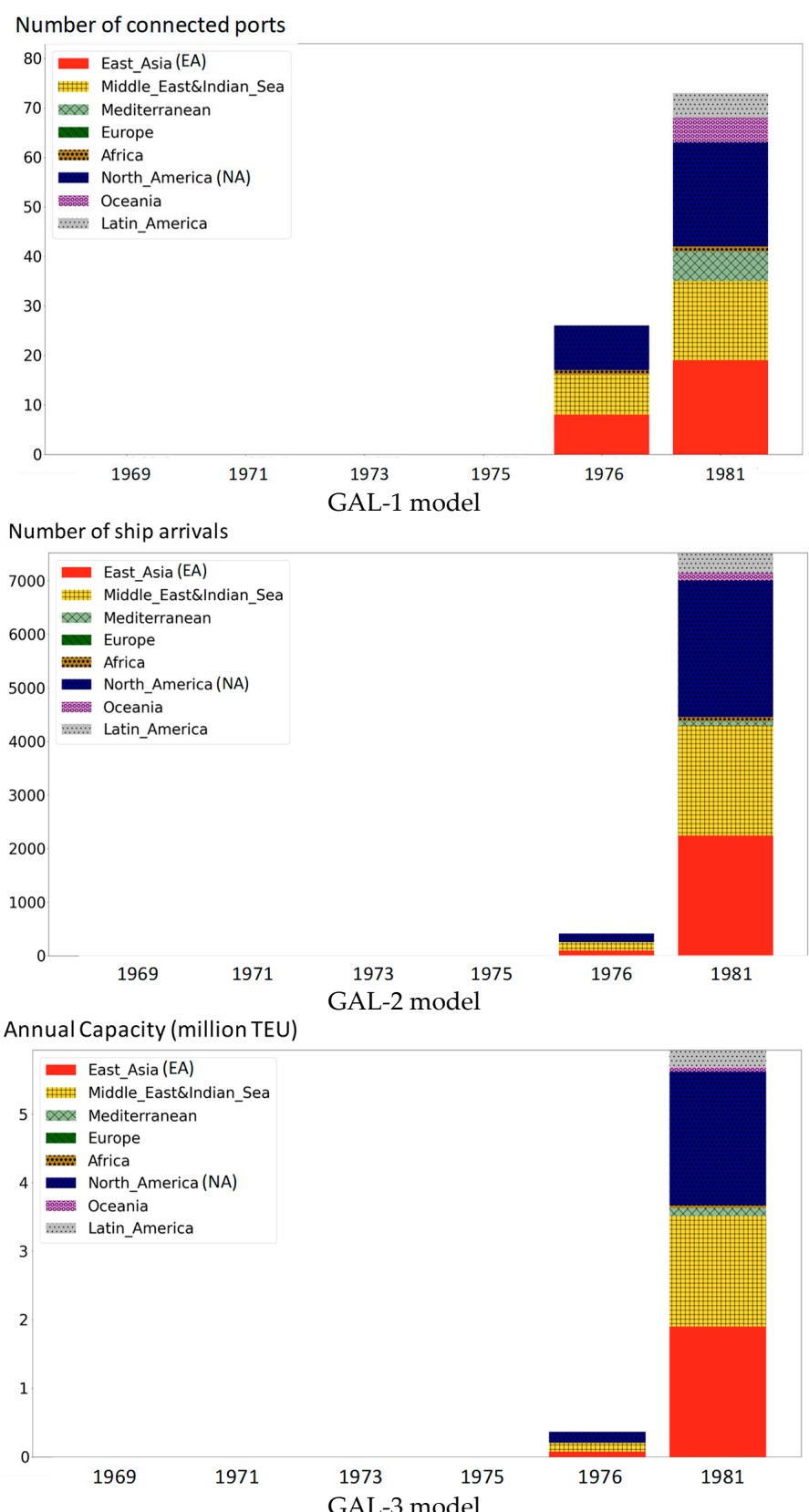

**Figure 14.** (Weighted) number of connected ports and regional breakdown in the port of Dubai (1969–1981).

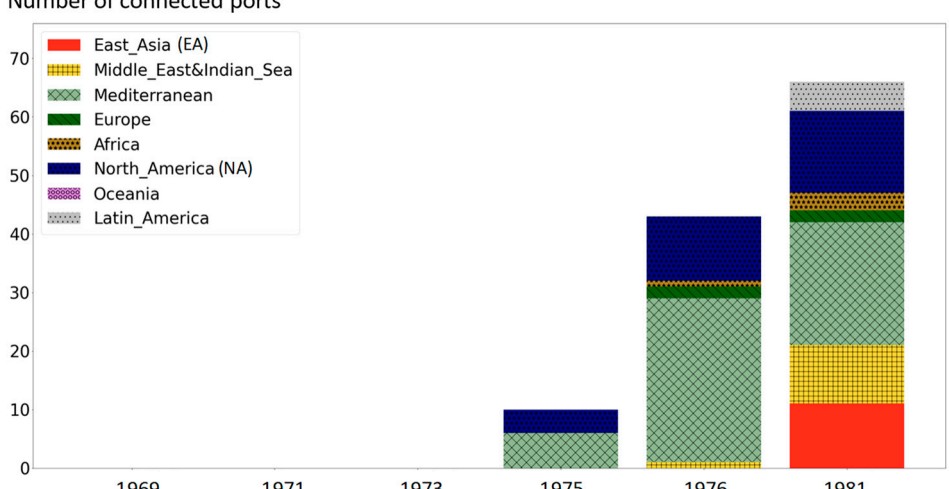

**Figure 15.** Number of connected ports and regional breakdown in the port of Alexandria (GAL-1 model, 1969–1981).

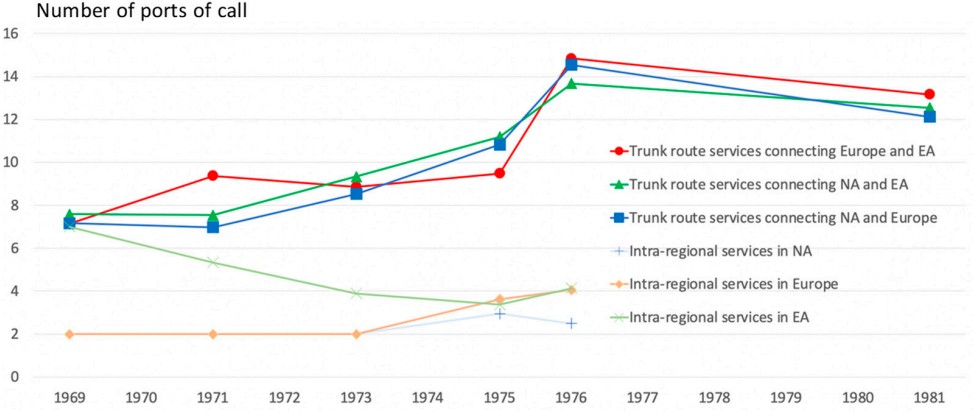

**Figure 16.** Average number of ports of call per service (1969–1981).

## 7. Conclusions

Using the global LS network data mainly provided by NYK, this study applied a network analysis method based on graph theory to the global MCS network, mainly in the 1970s. This study obtained a long-term overview that the network centered on NA, in the era of the emergence of MCS, had become multipolar by the 1980s and finally changed to a hub-and-spoke structure. Moreover, the importance of each port in the hub-and-spoke network was confirmed, and more elaborate and detailed clusters were detected using several models in which the method of edge construction and the consideration of frequency and capacity of each service are different.

Subsequently, through analyses focusing on the 1970s, changes in the MCS network were observed. These included the replacement of conventional LSs with MCS in trunk routes in the early 1970s and the development of the feeder transport network and the networks in the peripheral regions since the middle of the 1970s. There had been a relative increase in the number of ports in peripheral regions in the middle of the 1970s. These findings correspond with the descriptions in the NYK reports and Rua [34].

Moreover, detailed analyses focusing on the reopening of the SC in 1975 revealed discontinuous changes in inter-regional density from 1975 to 1976. Through both quantitative and qualitative discussions, it was found that the recession caused by the first oil crisis in 1973 decreased the MCS demand, whereas the reopening of the SC caused a surplus of containerships. Therefore, the number of ports of call increased, especially for the inter-regional LSs, which caused them to geographically limit the first and last ports

in each region to some specific ports, resulting in a decrease in inter-regional density. In conclusion, we can say that the reopening of the SC and the first oil crisis indirectly affected the global MCS network through the surplus of vessels rather than directly affecting it in a geographical sense.

We believe that this study contributes to accumulating the methodology and empirical knowledge on the vulnerability analysis of the present and future MCS networks, including the impact of shortcutting the shipping route for containership supplies. We used two different models (GDL and GAL) in terms of edge construction with three different types of edge weights according to the objective of the analysis and conducted a detailed analysis focusing on port characteristics in each region. The GDL is more suitable for the analysis of the movement network of containerships such as the first and last ports of call in the region and the shape of the networks through visualization, whereas the GAL is suitable for the analysis of hub-and-spoke structures with transshipments but not suitable for the visualization analysis because the density of the network is too high. Moreover, we confirmed the necessity of considering the indirect effects (such as economic downturn) of a wide range of events in the same era when analyzing the effects on the MCS network. This implication that the indirect effects should be sufficiently considered would be useful in predicting not only the effects when the SC is closed for a much longer period than the closure in 2021, but also the impacts of other types of current crises on network vulnerability for MCS, such as the heavy port congestion observed in the latter half of 2021 in the United States and the invasion of Ukraine by Russia.

However, owing to poor data accuracy, we were unable to analyze the MCS networks at shorter time intervals in the latter half of the 1970s and in the 1980s, in which a hub-and-spoke structure was formed. Therefore, more detailed and multifaceted analyses using more comprehensive data on longer time scales (for example, by digitizing Lloyd's data) should be conducted. The use of physical distance as the weight of the edges is also a further challenge, which is difficult to acquire exhaustively for all combinations of ports.

**Author Contributions:** Conceptualization, T.S. and R.S.; methodology, T.S. and S.M.; software, T.S. and S.M.; validation, T.S.; formal analysis, T.S.; investigation, T.S.; resources, T.M.; data curation, T.S., R.S. and T.M.; writing—original draft preparation, T.S. and R.S.; writing—review and editing, R.S. and K.T.; visualization, T.S. and S.M.; supervision, R.S.; project administration, R.S.; funding acquisition, K.T. All authors have read and agreed to the published version of the manuscript.

**Funding:** This research was funded by JSPS-18KK0051 and JSPS-20H00286.

**Institutional Review Board Statement:** Not applicable.

**Informed Consent Statement:** Not applicable.

**Data Availability Statement:** Not applicable.

**Conflicts of Interest:** The authors declare no conflict of interest.

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
