# Peer review of "Global Maritime Container Shipping Networks 1969–1981: Emergence of Container Shipping and Reopening of the Suez Canal"

_jmse, doi:10.3390/jmse10050602_

Round 1
Reviewer 1 Report
The paper is interesting but I would be interesting in the future if authors could compare their findings with present situation of container networks. This would than have greater influence towards accumulating the empirical knowledge about container networks.
Author Response
Thank you very much for your valuable comment. We appreciate that.
Reviewer 2 Report
Authors have spent great efforts to collect historical data published by the NYK shipping companies to review the impact of Suez Canal closure on the liner shipping networks development at that time and immediately after the canal is reopened. This research has its merits, but there are many issues to be clarified and to be revised. My specific comments for authors to make revisions are listed as followings.
- The major contribution of this manuscript is to “accumulating the empirical knowledge on the vulnerability analysis of the present and future maritime container shipping networks.” However, the authors used the 2016 data to exhibit the Figure 2 which cannot justify the goal shown in the abstract section. Why the 2016 data can be used to increase our knowledge on the present and future maritime container shipping networks?
- The definitions on GDL and GAL discussed in the section 3.3 are not clear. Although the authors have indicated GAL focuses on the connection between the LSs, why the GAL is suitable for the analysis of hub-and-spoke structures with transshipments? I cannot understand what “the connection between the liner shippings” is. Is the GAL composed of the links among all container ports even there are no direct service between some ports? It needs a more precise and clear definition.
- Definition on the table 1 is not clear. Why use the shipping frequency and annual shipping capacity as the edge weight? Is the shipping frequency measured on the annual basis as well? In section 3.2, the authors define the density, density within each community, and densities between different communities to award the Edge Weight. It seems capacity and frequency are highly correlated factors, is it appropriate to use the gap between model 2 and model 1/ model 3 to comes out the conclusion: “This result implies that the GDL-2 model can distinguish between ports connected by trunk routes of global MCS provided by larger vessels with a relatively lower frequency and those connected by feeder transports provided by smaller vessels with relatively higher frequency”.
- I don’t understand how Louvain method (a type of greedy optimization method) is used to estimate the communities, it is not discussed in details.
- Figure 4 is not provided in the main text.
- Typos are found in most of the figures. “Newyork” should be the typo for “New York.” Corrections are required for “Lehavre” and “Hongkong” in the figure 5 and figure 7.
- The score shown in the Figure 3 indicates “European ports took the lead in 1981, and the difference in the betweenness centrality between the three major regions was also reduced.” How the score are obtained remained not clearly explained. Is it calculated by the network adjacency matrix discussed in the page 5?
- Why “the ‘major’ community is defined as the one that occupies 2 per cent or more of the total number of ports at each time point.”?
- Figure 11, the wordings “GAL-2 model” is not correctly positioned. The shaded area of the Figure 11 cannot be clearly differentiated if the journal is printed in the black and white colour.
- “The inter-regional density in the GDL-1 model, therefore, declined in 1976 because the increase in the number of ports of call in the inter-regional LSs geographically limited the first and last ports of these regions to some specific ports such as Singapore and Antwerp.” Is Singapore the last port of call in the Asian region? If it is so, then the wordings should be revised as following “The inter-regional density in the GDL-1 model, therefore, declined in 1976 because the increase in the number of ports of call in the inter-regional LSs geographically limited the first and last ports of these regions to some specific ports such as Antwerp and Singapore.”
- The authors argue “The use of physical distance as the weight of the edges is also a further challenge.” Why don’t the author do so in this research?
Author Response
Thank you for your valuable comments. Our responses are as follows.
1. Figure 3 (in the new version of the paper. It was Figures 2 in the old version) was shown for the visualization of different graph methods of networks. The choice of year was taken by the latest availability of the series of data, which can best describe the differences in the networks. We inserted one phase in the last para of section 3.3 to explain that. As described in the last para of section 1, the entire history of the development of the global MCS network since its emergence of MCS in the 1970s to 2016 was analysed in section 4 for giving broader perspective to place our analysis before the following analyses in section 5 and 6 that are our main focus in this paper; therefore, this is not included in the abstract.
2. The last sentence of first para in section 3.3 was revised according to the comment. Also, we corrected keywords that “LS” should stand for “liner service”, not “liner shipping.”
3. Yes, GDL-2 model consider the annual number of vessel movements. GDL-3 model consider the annual vessel capacities which are acquired by multiplying annual frequency by average capacity per vessel. The second para of section 3.3 is revised accordingly. Also we are very sorry that the results of GDL-2 and GDL-3 models shown in Table 2 were opposite. We corrected them as well as the discussion part in the manuscript (i.e. last two paras of section 4).
4. We added some explanations accordingly in the last para of section 3.2.
5. We are very sorry. It was added (as Figure 5).
6. We revised Figures 3, 6, and 8 (in the new version of the paper. They were Figures 2, 5, and 7 in the old version) accordingly. Please note that the shape of each figure is slightly changed due to the upgrade of the software, although the input data and analysis results are not changed.
7. This implication is directly acquired from Figure 4 (in the new version of the paper. It was Figure 3 in the old version) when we focus on the major ports in each major region. We corrected the last sentence of second para in section 5 accordingly.
8. We added the explanation in the last para of section 4.
9. The title of each figure in Figure 12 (in the new version of the paper. It was Figure 11 in the old version) is corrected. The authors understand all figures are electronically displayed with colours.
10. We corrected accordingly.
11. It is mainly due to data availability. We added the explanation in the last sentence of the paper.
Reviewer 3 Report
I’m glad to learn knowledge from this paper and value the authors’ work.The story is interesting and I’m attracted by this paper and find some problems in this paper:
1.This study applied graph theory to conduct an empirical analysis of the evolution of global maritime container shipping networks, mainly focusing on the 1970s. In addition to analysing the change in overall structures of the networks over the long term (from the 1970s to the present) andmid-term (in the 1970s).”We use a data source (World Container Fleets and Their Services: 日本郵船調査部編『世界のコンテナ船隊および就航状況』) published by Nippon Yusen Kaisha (hereafter, the NYK report) for developing liner service (LS) network data of global MCS, mainly in the 1970s”.As I know,some reference papers in this paper use AIS to exact OD data source.We have little knowledge in 日本郵船調査部編『世界のコンテナ船隊および就航状況』.Can you offer more detail and advantages of your data source in this paper?
2.Direct linkages (GDL) and a graph of all linkages (GAL) is dominated in this paper.The paper said”In contrast, GAL connects with edges all ports that are called to in the same LS. The GDL is suitable for the analysis of the movement network of container ships”.This sentence must be proved effective by other reference papers.
3.The figures in this paper used by Gephi software.But I think picture quality is worrying.The ports in their communities were indistinguishable.
4.Figure 4 is missing.
5.Because of quality of Figure5,”The following analyses in this section focus on the MCS network at four time points.......”can’t be proved by figures.
6.In section 4,the paper said ”This result implies that the GDL-2 model can distinguish between ports connected by trunk routes of global MCS provided by larger vessels with a relativelylower frequency and those connected by feeder transports provided by smaller vessels with relatively higher frequency. ”However,the analysis was mainly based on Number of major communities.The argument is slightly insufficient and we encourage you to analyze it with conditions of regions and ports.
7. In figure 9,I think the number of ports is fewer in the picture for showing classification of port groups.
Author Response
I’m glad to learn knowledge from this paper and value the authors’ work. The story is interesting and I’m attracted by this paper and find some problems in this paper:
- This study applied graph theory to conduct an empirical analysis of the evolution of global maritime container shipping networks, mainly focusing on the 1970s. In addition to analysing the change in overall structures of the networks over the long term (from the 1970s to the present) and mid-term (in the 1970s).” We use a data source (World Container Fleets and Their Services: 日本郵船調査部編『世界のコンテナ船隊および就航状況』) published by Nippon Yusen Kaisha (hereafter, the NYK report) for developing liner service (LS) network data of global MCS, mainly in the 1970s”. As I know, some reference papers in this paper use AIS to exact OD data source. We have little knowledge in 日本郵船調査部編『世界のコンテナ船隊および就航状況』.Can you offer more detail and advantages of your data source in this paper?
→ Each item included in the NYK report was explained in the first para of section 3.1. We also added a sample copy of the NYK report as a new figure (Figure 1).
2.Direct linkages (GDL) and a graph of all linkages (GAL) is dominated in this paper. The paper said “In contrast, GAL connects with edges all ports that are called to in the same LS. The GDL is suitable for the analysis of the movement network of container ships”. This sentence must be proved effective by other reference papers.
→ Our understanding on the difference between GDL and GAL is slightly different from the past papers. To clarify our understanding more, we added several phrases and words in the first and last para of section 3.3 and in the fourth para of section 7.
- The figures in this paper used by Gephi software. But I think picture quality is worrying. The ports in their communities were indistinguishable.
→ We revised Figures 3, 6, and 8 (in the new version of the paper. They were Figures 2, 5, and 7 in the old version) to increase the resolution of each figure and the visibility of each port name. However, some duplications in port names still remain because they are preset by the software and cannot be manually modified. Also, please note that the shape of each figure is slightly changed due to the upgrade of the software, although the input data and analysis results are not changed.
4.Figure 4 is missing.
→ We are very sorry for missing it. It was added.
- Because of quality of Figure 5, “The following analyses in this section focus on the MCS network at four time points.......” can’t be proved by figures.
→ As written in the response against the third comment, we revised Figures 3 and 6 (in the new version of the paper. They were Figures 2 and 5 in the old version. Same as below) to increase the resolution of each figure. In addition, we’d like to emphasize that the latter half of the discussions in this para are based on Figure 4 (Figure 3), not Figures 3 nor 6 (Figures 2 nor 5). We only discussed the rough historical changes in the shape of global container shipping network from Figures 3 and 6 (Figures 2 and 5), which can be sufficiently confirmed from these figures.
- In section 4, the paper said “This result implies that the GDL-2 model can distinguish between ports connected by trunk routes of global MCS provided by larger vessels with a relatively lower frequency and those connected by feeder transports provided by smaller vessels with relatively higher frequency.” However, the analysis was mainly based on Number of major communities. The argument is slightly insufficient and we encourage you to analyze it with conditions of regions and ports.
→ We added another related result (Figure 7) on the list of Top 20 ports in betweenness centrality and their scores estimated from GDL-2 and GDL-3 models and discussion about it, according to your advice.
- In figure 9, I think the number of ports is fewer in the picture for showing classification of port groups.
→ The ports shown in Figure 10 (Figure 9) are representative ports in each group. We added this explanation in the title of Figure 10.
Reviewer 4 Report
This paper is of sound quality on a subject deserving the Journal's attention. Applying graph theory, this study empirically analyzed the evolution of global maritime container shipping networks focused on the 1970s. Research results identified that the initial single polar network structure, in which New York and other North American ports were placed at the center, changed to a multipolar structure, finally forming a hub-and-spoke structure. Overall, the paper is well written and well structured, therefore it is easy to follow and builds a clear conclusion from the data. Generally well written but requires some editing and revision.
Additionally, this study clearly presented the finding of this study, but research implication part is weak. A few sentences describe practical implications. Research conclusion (implication) part is weak, focusing on data analysis (enumerate bits of information). Additional explanations incorporating theoretical and practical are required.
Author Response
Thank you very much for your valuable comments. We corrected our manuscript according to the comments provided by all reviewers.
Also, we added the several sentences and phrases in the latter part of the conclusion section according to your advice.
Round 2
Reviewer 2 Report
I don't have any further comment. All of my previous comments are successfully answered/revised by the authors.